# Query-based Adversarial Attacks on Graph with Fake Nodes

**Zhengyi Wang** [1]  **Zhongkai Hao** [2]  **Jun Zhu** [1]

## Abstract

While deep neural networks have achieved great success on the graph data analysis, recent works have shown that they are vulnerable to adversarial attacks where fraudulent users can fool the model with a limited number of queries. Compared with adversarial attacks on image classification, performing adversarial attack on graphs is challenging because of the discrete and non-differential nature of a graph. To address these issues, we proposed Cluster Attack, a novel adversarial attack by introducing a set of fake nodes to the original graph which can mislead the classification on certain victim nodes. Moreover, our attack is performed in a practical and unnoticeable manner. Extensive experiments demonstrate the effectiveness of our method in terms of the success rate of attack.

## 1. Introduction

Recent research in Graph Neural Networks (GNNs) has shown a promising performance on various applications to graph data including the recommender systems (Ying et al., 2018), social networks (Qiu et al., 2018), electronic commerce (Chen et al., 2019), etc. Just like other types of deep learning models, recent studies have shown that GNNs are vulnerable to adversarial attack (Dai et al., 2018; Zügner et al., 2018). The performance of a well-trained GNN can be significantly degenerated by adversarial manipulations, which are carefully crafted inputs with small perturbations added.

In this work, we consider a more practical scenario in adversarial attack on graph data, which aims to mislead the predicted labels of certain victim nodes without sacrificing the prediction on other nodes significantly. In our setting,

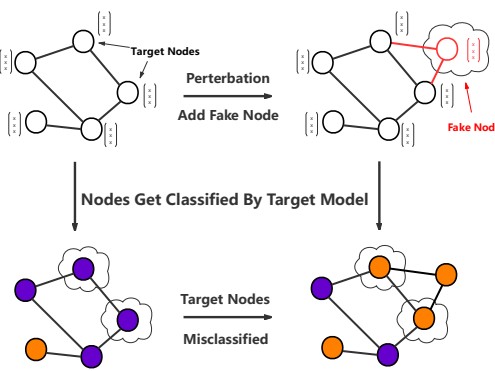

*Figure 1.* An Illustration of Fake Node Attack

we consider to protect the neighbors of victim nodes within $k$-hop from being misclassified. Moreover, we choose to introduce extra fake nodes into the original graph instead of directly modifying the original graph, since it is more practical in real-world scenario.

It is also noted that in the practical scenarios, we can only access part of the graph. In our setting, we can only access *partial information* of the graph data by observe the nodes being attacked with their neighbor nodes within $k$-hop. As a more challenging setup, we have no access to to the model parameters. We can only have a limited number of queries on the victim model about the predicted scores of certain nodes when performing adversarial attack.

To tackle the discrete optimization problem, we propose our *Cluster Attack*, which attacks by divide the victim nodes into several clusters according to their *most adversarial feature*s, which can be approximately computed within limited queries.

Our contribution can be summarized as follows:

- We propose a practical threat model on graph adversarial attack. We perform query-based adversarial attack on graph with partial information about the graph with non-targeted nodes protected.

- We propose *Cluster Attack*, an effective and efficient adversarial attack on graph structured data which attacks by divide victim nodes into several smaller clusters.

---

[1]Department of Computer Science and Technology, Tsinghua University, China [2]University of Science and Technology of China, China. Correspondence to: Zhengyi Wang <coolboy-wzy@gmail.com>.

*Accepted by the ICML 2021 workshop on A Blessing in Disguise: The Prospects and Perils of Adversarial Machine Learning.* Copyright 2021 by the author(s).

## 2. Related Works

In recent years, many methods were proposed to perform adversarial attack on graph. For greedy methods, NET-TACK (Zügner et al., 2018) proposed to attack the graph by greedily modifying the edges and features of the graph. (Wang et al., 2018) proposed to attack by adding fake nodes and then greedily modifying the edges and features. For methods based on reinforcement learning, (Dai et al., 2018) proposed RL-S2V to attack by changing existing edges in the graph, while (Sun et al., 2020) proposed NIPA to attack by adding fake nodes. (Xu et al., 2019; Wu et al., 2019; Chen et al., 2020b;a) proposed to attack using gradient information. (Zügner & Günnemann, 2019) uses meta-learning to attack. (Ma et al., 2019) attacks by Rewiring, a special operation on the graph. (Chang et al., 2019) proposed GF-Attack, a restricted black-box adversarial framework. (Ma et al., 2020) proposed a black-box attack strategy manipulating the original graph. (Xu et al., 2020) works contemporarily with us which performs white-box attack by adding fake nodes. The above methods are either not adoptable in our setting or having poor performance in our setting.

## 3. Methodology

### 3.1. Problem Formulation

Given a set of victim nodes $\Phi_\mathbf{A} \subseteq \Phi$ in the graph, our goal is to perform mild perturbations on the graph $G = (\mathbf{A}, \mathbf{X})$, leading to $G^+ = (\mathbf{A}^+, \mathbf{X}^+)$, such that the predicted labels of as many nodes as possible in $\Phi_\mathbf{A}$ change. $\mathbf{A}^+ = \begin{bmatrix} \mathbf{A} & \mathbf{B}^T \\ \mathbf{B} & \mathbf{A}_{fake} \end{bmatrix}$ and $\mathbf{X}^+ = \begin{bmatrix} \mathbf{X} \\ \mathbf{X}_{fake} \end{bmatrix}$, where $\mathbf{B}$ represents the connections between original nodes and fake nodes. $\Phi^+ = \Phi \cup \Phi_{fake}$ denotes the node set of $G^+$. Starting from $\mathbf{A}_{fake} = \mathbf{0}, \mathbf{B} = \mathbf{0}$, we manipulate $\mathbf{A}_{fake}, \mathbf{B}$ and $\mathbf{X}_{fake}$, leading to as low classification accuracy on $\Phi_\mathbf{A}$ as possible.

Our **thread model** is defined as follows.

**Adversarial Budget** To ensure our perturbation is unnoticeable, we limit the number of new connections between fake nodes and original nodes by $\Delta_{edge}$. The connections between fake nodes, i.e., $\mathbf{A}_{fake}$, are free for us to modify while the connections between original nodes, i.e., $\mathbf{A}$, are not allowed for us to modify. We can decide $\mathbf{X}_{fake}$ at will but cannot change $\mathbf{X}$, the features of original nodes. We limit the number of fake node by the number of rows of $\mathbf{A}_{fake}$.

**Protected Nodes** Owing to the non-i.i.d nature of graph data, attacking victim nodes may have side effects on their neighboring nodes which we are not aimed for. While attacking victim nodes, we aim to keep the labels of other nodes which are not targeted unchanged at the same time to

make our perturbation unnoticeable. In our setting, we consider to protect $\mathcal{N}_k(\Phi_\mathbf{A})$, neighbors of victim nodes within $k$-hop from being misclassified.

**Partial Information** We can only query the classification results of the victim nodes with their neighbors within $k$-hop and fake nodes. We can only make connections between victim nodes and fake nodes.

**Limited Queries** We can totally query $K$ times for the predicted scores of all victim nodes with their neighbors within $k$-hop. The architecture and parameters about victim model are unknown by the attacker.

We aim to make the classifier misclassify as many nodes in $\Phi_\mathbf{A}$ as possible. We formulate our problem as an optimization problem. Directly optimizing the number of misclassified nodes is difficult as the objective is discrete. Thus we try to optimize a substitute loss function as

$$\min_{G^+} \mathbb{L}(G^+; \Phi_\mathbf{A}) \triangleq \sum_{v \in \Phi_\mathbf{A}} \ell(G^+, v) + \lambda \sum_{v \in \mathcal{N}_k(\Phi_\mathbf{A})} \ell_\mathcal{N}(G^+, v),$$

(1)

$$s.t. \; \mathbb{N}_r(G^+) \le N_{fake}, \; \mathbb{N}_e(G^+) \le \Delta_{edge},$$

where $G^+ = (\begin{bmatrix} \mathbf{A} & \mathbf{B}^T \\ \mathbf{B} & \mathbf{A}_{fake} \end{bmatrix}, \begin{bmatrix} \mathbf{X} \\ \mathbf{X}_{fake} \end{bmatrix})$. $\mathbb{N}_r(G^+)$ denotes the number of rows of matrix $\mathbf{A}_{fake}$ and is no more than $N_{fake}$, which means that we at most introduce $N_{fake}$ fake nodes into the original graph. $\mathbb{N}_e(G^+)$ represents the number of non-zero elements of $\mathbf{B}$ and is no more than $\Delta_{edge}$, which means that we can at most add $\Delta_{edge}$ extra links. $\ell(G^+, v)$ and $\ell_\mathcal{N}(G^+, v)$ represents loss function for every victim node and every protected node, respectively. Smaller $\ell(G^+, v)$ means node $v$ is more likely to be misclassified by victim model $f$ and smaller $\ell_\mathcal{N}(G^+, v)$ means the predicted label of node $v$ is less likely to be changed during our attack. We perform targeted attack, which means the labels of victim nodes have to be misclassified as the ones which we specify.

Here we choose the C&W loss (Carlini & Wagner, 2016)

$$\ell(G^+, v) = \max(\max_{y_i \neq y_t}([f(G^+)]_{v,y_i}) - [f(G^+)]_{v,y_t}, 0),$$

(2)

for our attack, where $y_t$ stands for the target label of node $v$ and the attacker succeeds only when node $v$ is misclassified as $y_t$. $[f(G^+)]_{v,y_i}$ denotes the output value of node $v$ having label $y_i$. For protected nodes, we have

$$\ell_\mathcal{N}(G^+, v) = \max(\max_{y_i \neq y_g}([f(G^+)]_{v,y_i}) - [f(G^+)]_{v,y_g}, 0),$$

(3)

where $y_g$ stands for the ground-truth label of node $v$ provided by victim model. Overall loss $\mathbb{L}(G^+; \Phi_A)$ is summed over the loss of each victim node along with the loss of each

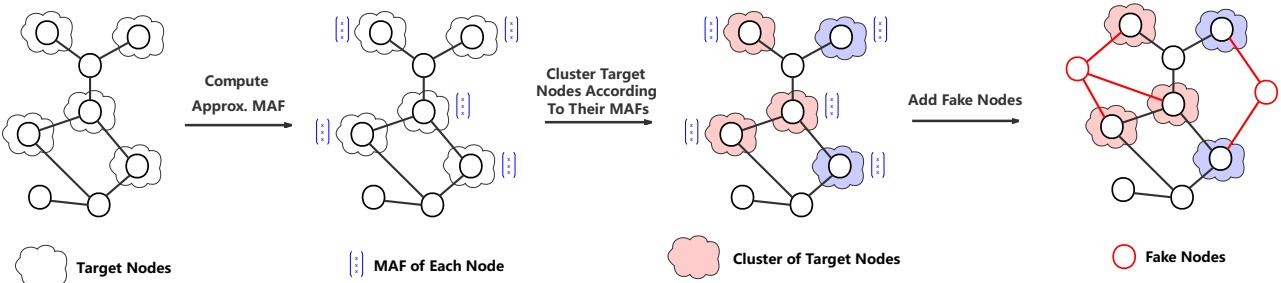

*Figure 2.* An Illustration of Our Cluster Attack

protected neighboring node with a trade-off parameter $\lambda$. In practice, we add square root to the loss of each victim node to favor the nodes which are likely to be misclassified.

Optimization of Eq. (1) is challenging due to the discrete nature of $G^+$ and large space of possible choices of $G^+$. To tackle the optimization problem, we propose our Cluster Attack.

### 3.2. Cluster Attack

In an adversarial scenario, it is often the case that the number of fake nodes we are allowed to add to the graph is much smaller than the number of victim nodes. To make use of each fake node better, we have to connect every fake node to several victim nodes. However, due to the structural complexity of the graph, different victim nodes may have very different local structures and the corresponding feature information, especially when our victim nodes are sparsely scattered in the whole graph. Consequently, a fake node with certain feature may change the predicted label of one victim node after connecting to it, but may not change another victim node's.

Building on the above perspective, we have the insight that, if we connect a fake node to several victim nodes which share a similarity that their predicted labels are all easily changed after they are connected to fake nodes with similar features, then we have more chance of changing the predicted labels of those victim nodes. As a result, we can divide the victim nodes into several clusters according to this similarity and for each cluster we assign one fake node to attack.

**Most Adversarial Feature**   In our attack, we use the *most adversarial feature*, which represents the vulnerability of each victim node, to denote the above similarity of victim nodes. It's the feature of the fake node connected to the victim node which minimizes our loss.

Formally, for victim node $v$, consider that we only connect one fake node to and only to $v$. (To achieve this, we may

temporarily remove all the extra edges in $G^+$ and temporarily add one edge connecting this victim node and one fake node $v_f$. After the computation, the edges in $G^+$ become the same before the computation.) $x_{v_f}$ is the feature of the newly added fake node $v_f$. The MAF of $v$ is defined as followed

$$\text{MAF}(v) = \underset{x_{v_f}}{\operatorname{argmin}} \ \mathbb{L}(G^+; \Phi_{\mathbf{A}}), \qquad (4)$$

where $\mathbb{L}(G^+; \Phi_{\mathbf{A}})$ is defined in Eq. (1). The MAF is related to the victim node's gradient towards adversarial examples.

Direct computation of MAF for every victim node may be extremely time-consuming and intractable with only a limited number of queries. Here we propose an greedy algorithm to approximate $\text{MAF}(v)$ with $K_t$ queries for each victim node which is summarized in Algorithm 1. For convenience and consistency, we use the same notation $\text{MAF}(v)$ to denote approximated MAF of node $v$ computed using Algorithm 1 in the following sections.

---

**Algorithm 1** Fast Approximation of MAF with a Fixed Number of Queries

---

**Input:** Graph $G^+ = (\mathbf{A}^+, \mathbf{X}^+)$. Victim node $v$. Number of queries $K_t$.

**Output:** Approximated most adversarial feature $\text{MAF}(v)$ for $v$

1: **initialize** Choose one fake node $v_f$ and connect it to and only to $v$, randomly initialize the fake node's feature $x_{v_f}$. Keep other fake nodes isolated.

2: Randomly sample a sequence $I_t$ from $\{1, 2, ..., D\}$ with length $K_t$. $D$ is the dimension of nodes' feature.

3: **for** $i \in I_t$ **do**

4:   **if** $x_{v_f}[i] \leftarrow 1 - x_{v_f}[i]$ makes $\mathbb{L}(G^+; \Phi_{\mathbf{A}})$ smaller **then**

5:     $x_{v_f}[i] \leftarrow 1 - x_{v_f}[i]$

6:   **end if**

7: **end for**

8: **return** $\text{MAF}(v) \leftarrow x_{v_f}$

---

*Table 1.* Success rates of Cluster Attack along with other baselines. $T$ denotes number of victim nodes. For $T = 3$, we only add 3 fake nodes in our Cluster Attack.

| Method | Cora | | | | Citeseer | | | |
|---|---|---|---|---|---|---|---|---|
| | $T = 3$ | $T = 5$ | $T = 7$ | $T = 10$ | $T = 3$ | $T = 5$ | $T = 7$ | $T = 10$ |
| Random | 0.07 | 0.08 | 0.04 | 0.05 | 0.04 | 0.02 | 0.03 | 0.03 |
| NETTACK | 0.61 | 0.57 | 0.55 | 0.53 | 0.75 | 0.71 | 0.66 | 0.61 |
| NETTACK - Sequential | 0.68 | 0.73 | 0.72 | 0.70 | 0.76 | 0.74 | 0.72 | 0.67 |
| Fake Node Attack | 0.61 | 0.58 | 0.54 | 0.52 | 0.76 | 0.68 | 0.62 | 0.60 |
| KDD Cup 1st Attack | 0.61 | 0.55 | 0.51 | 0.42 | 0.55 | 0.56 | 0.51 | 0.45 |
| **Cluster Attack** | **0.99** | **0.93** | **0.84** | **0.72** | **1.00** | **0.89** | **0.80** | **0.70** |

**Clustering the Victim Nodes** After the computation of most adversarial features, we divide the victim nodes $\Phi_{\mathbf{A}}$ into $N_{fake}$ clusters $C = \{C_1, C_2, ..., C_{N_{fake}}\}$ according to their MAFs using K-Means algorithm.

**Overall Algorithm** Figure 2 is an overview of our attack algorithm. We only consider the case that $\Delta_{edge} \geq |\Phi_{\mathbf{A}}|$, otherwise some victim nodes do not have the chance to connect directly to fake nodes. For now, we keep $\mathbf{A}_{fake} = \mathbf{0}$ during our whole algorithm. We leave utilizing $\mathbf{A}_{fake}$, the connections between fake nodes, to enhance our attack as our future work. Our method prevents the time-consuming search of the large space of $\mathbf{A}_{fake}, \mathbf{B}, \mathbf{X}_{fake}$ and is thus an efficient method.

## 4. Experiments

### 4.1. Experimental Setup

We do our experiments on Cora and Citeseer(Sen et al., 2008), two benchmark citation networks. The statistics of the datasets are shown in Table 2.

*Table 2.* Statistics of the datasets

| Name | Nodes | Edges | Features | Classes |
|---|---|---|---|---|
| Cora | 2708 | 5429 | 1433 | 7 |
| Citeseer | 3327 | 4732 | 3702 | 6 |

For each experimental setting, we run 100 times of experiments and report the average results. Each round we randomly sample $|\Phi_A|$ nodes in the graph as victim nodes. To reduce the variance in the training process of victim model, we retrain the victim GCN model every 5 rounds of attack. Number of queries $K$ is set to $K = |\Phi_A| \cdot K_t + N_{fake} \cdot K_f$, where we set $K_t = D$ and $K_f = D$. $D$ is the dimension of node's feature. We set $k = 1$ in $\mathcal{N}_k(\Phi_{\mathbb{A}})$, which means we can only observe 1-hop neighbors of victim nodes and we aim to protect those 1-hop neighbors. By default we set trade-off parameter $\lambda = 0$ without specification.

We compare our method with baselines including Random Attack, NETTACK (Zügner et al., 2018), NETTACK - Sequential proposed by us which adds fake nodes sequentially, Fake Node Attack (Wang et al., 2018) and the method which won 1st place in KDD Cup competition on graph adversarial attack (kdd, 2020). For baselines, we don't limit the number of queries.

### 4.2. Performance with Different Number of Victim Nodes

We evaluate the performance of Cluster Attack along with other baselines with different number of victim nodes. Without loss of generality, we uniformly set $N_{fake} = 4$, $\Delta_{edge} = |\Phi_A|$ and let the number of victim nodes varies to see the performance under different $N_{fake} : |\Phi_A|$. We compare Cluster Attack with other baselines. The results are shown in Table 1. Our algorithm outperforms all baselines in terms of success rates.

### 4.3. Other Experiments

Other experiments demonstrating the effectiveness of our Cluster Attack are shown in appendix, including experiments with different number of fake nodes, experiments with different trade-off parameter, experiments with different number of queries, analysis of cluster attack on victim nodes with different degrees and ablation study.

## 5. Conclusion

In this paper, we propose Cluster Attack, an algorithm of adversarial attack on graph structured data. We perform query-based black-box adversarial attack on graph by adding fake nodes with partial information about the graph. We further consider to protect the predicted labels of neighboring nodes of victim nodes from being changed. We propose to attack by clustering the victim nodes according to the similarity in their most adversarial features, which can be approximated by a limited number of queries. Experimental results demonstrate our method has strong performance.

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

# A. Supplemental Material

## A.1. Performance with Different Number of Fake Nodes.

In this section, we evaluate the performance of Cluster Attack along with other baselines with different number of fake nodes. We fix the number of victim nodes at 10 and vary the number of fake nodes to examine the success rates. We set $\Delta_{edge} = |\Phi_A| = 10$. The success rates are shown in Figure 3. We only list the results of NETTACK - Sequential in Figure 3 without NETTACK since we found that NET-TACK - Sequential performs better than NETTACK. We see that the success rate is higher when there are more fake nodes. For Cluster Attack, we conjecture that this is because the number of clusters get larger when there are more fake nodes. Thus the MAFs of the victim nodes in each cluster are able to be closer to each other and they are easier to be attacked by the same fake node. Among all methods, our Cluster Attack achieves the highest success rate.

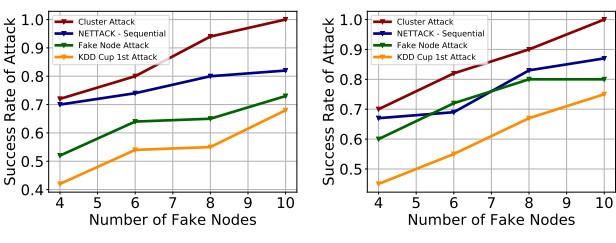

(a) Experiment on Cora     (b) Experiment on Citeseer

*Figure 3.* Success Rates of Cluster Attack with Different Number of Fake Nodes.

## A.2. Performance with Different Trade-Off Parameter $\lambda$.

In this section, we examine the performance of Cluster Attack with different trade-off parameter $\lambda$ between fake nodes and protected nodes. We examine the performance under different $\lambda$ in Cora dataset. We uniformly set $N_{fake} = 4$, $\Delta_{edge} = |\Phi_A| = 10$. We choose 2 most competitive baselines, NETTACK - Sequential and Fake Node Attack and adapt their loss function to our trade-off format. The results are shown in Figure 4. It can be seen from Figure 4 that our algorithm performs the best among all baselines in terms of attack success rates. When $\lambda$ goes up, which means that we pay more attention to the protected nodes, the percentage of protected nodes whose labels remain unchanged during attack goes up while the success rates of attack drops. When $\lambda$ gets large enough ($\lambda \geq 10$), nearly all protected nodes are successfully protected, which demonstrates the effectiveness of our trade-off formulation in our loss function Eq. (1). It shows that to protect the labels of not-targeted nodes from being changed during attack, we can simply set

a large $\lambda$. Also, our trade-off formulation between victim nodes and protected nodes in Eq. (1) not only applies to our Cluster Attack, but also applies to other baselines.

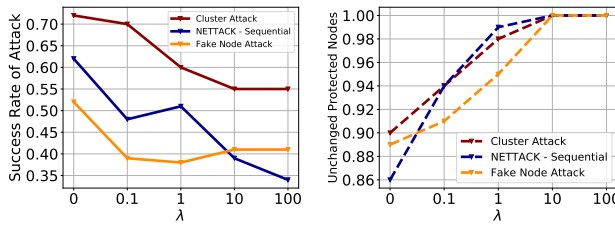

(a) Success Rates of Attack    (b) Percentage of Protected Nodes whose Labels Remain Unchanged during Attack

*Figure 4.* Cluster Attack along with Other Baselines in Cora with Different $\lambda$.

## A.3. Performance with Different Number of Queries.

In this section, we examine the performance of Cluster Attack with different number of queries. We set $K_t = K_f = \alpha \cdot D$ and examine the performance under different $\alpha$ in Cora and Citeseer dataset. We uniformly set $N_{fake} = 4$, $\Delta_{edge} = |\Phi_A| = 10$. The results are shown in Figure 5. The success rate of Cluster Attack drops as the number of queries drops. Our algorithm still performs well when the number of queries drops not very much, especially when $\alpha \geq 0.4$. It demonstrates that the most adversarial feature can be approximated and the fake nodes' features can be optimized with a smaller number of queries without a great decrease in the success rate. It shows that our Cluster Attack can work in a query-efficient manner.

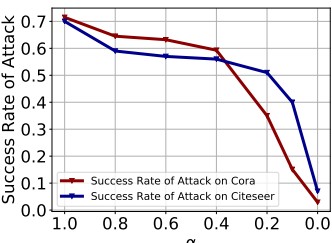

*Figure 5.* Success Rates of Cluster Attack with Different Number of Queries in Cora and Citeseer.

## A.4. Analysis of Cluster Attack on Victim Nodes with Different Degrees.

In this section, we evaluate the performance of Cluster Attack on victim nodes with different degrees. We uniformly set $N_{fake} = 4$, $\Delta_{edge} = |\Phi_A| = 10$. The success rates of Cluster Attack on victim nodes with different degrees are

shown in Figure 6 along with the proportion of sampled victim nodes with each degree. Victim nodes with degrees larger than or equal to 7 are counted together since they only account for a small proportion.

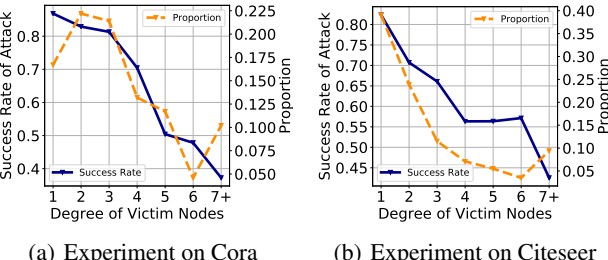

(a) Experiment on Cora      (b) Experiment on Citeseer

*Figure 6.* Success Rates of Cluster Attack on Victim Nodes with Different Degrees.

It can be seen from Figure 6 that victim nodes with higher degrees are more robust to our attack in general. We conjecture that when a victim node has a relatively large number of neighbors, adding one fake node as its neighbor has less impact on it and thus is less likely to change its predicted label.

### A.5. Ablation Study.

In this section, we examine the effectiveness of our most adversarial feature (MAF). We uniformly set $N_{fake} = 4$, $\Delta_{edge} = |\Phi_A| = 10$. We compare the success rate of Cluster Attack without MAF, i.e., the victim nodes' MAFs are randomly set. The results are shown in Table 3. Cluster Attack without MAF performs worse than normal Cluster Attack with MAF, which demonstrates the effectiveness of our MAF. MAF is related to the vulnerability of victim nodes. Nodes with similar MAFs in a cluster are easier to be affected together by one fake node. Thus the success rate of Cluster Attack with MAF is better than without MAF.

*Table 3.* Success Rates of Cluster Attack with and without MAF in Cora.

| Method | Success Rate |
|---|---|
| Cluster Attack - without MAF | 0.62 |
| Cluster Attack | **0.72** |