# OpenReview forum: "Query-based Adversarial Attacks on Graph with Fake Nodes"
_ICML.cc/2021/Workshop/AML — ICML 2021 Workshop AML Poster_

### Official Review · Reviewer_15NG · 2021-06-19
**A good method performing attack on graph, with solid math modeling and sufficient ablation study.**

**Rating:** Accept
**Confidence:** 3

**Review:**

Strength:
1) The English is good.
2) The model is clear. The experiments in Appendix are sufficient and meaningful to valid the effectiveness of the proposed Cluster Attack.

Weakness:
1) It will be better if the authors add more comparing experiments and analysis in Section 4.

---

### Decision · Program_Chairs · 2021-06-21

**Decision:**

Accept (Poster)

**Comment:**

This paper proposed a new attack method on graph with solid modeling and sufficient ablation study.